# Hypothetical Mechanism of Exercise-Induced Acute Kidney Injury Associated with Renal Hypouricemia

**DOI:** 10.3390/biomedicines9121847

**Published:** 2021-12-06

**Authors:** Makoto Hosoyamada

**Affiliations:** Department of Human Physiology and Pathology, Faculty of Pharma-Science, Teikyo University, 2-11-1 Kaga, Itabashi, Tokyo 173-8605, Japan; hosoyamd@pharm.teikyo-u.ac.jp

**Keywords:** renal hypouricemia, TLR4, NLRP3 inflammasome, IL-1β, EDHF

## Abstract

Renal hypouricemia (RHUC) is a hereditary disease that presents with increased renal urate clearance and hypouricemia due to genetic mutations in the urate transporter URAT1 or GLUT9 that reabsorbs urates in the renal proximal tubule. Exercise-induced acute kidney injury (EIAKI) is known to be a complication of renal hypouricemia. In the skeletal muscle of RHUC patients during exhaustive exercise, the decreased release of endothelial-derived hyperpolarization factor (EDHF) due to hypouricemia might cause the disturbance of exercise hyperemia, which might increase post-exercise urinary urate excretion. In the kidneys of RHUC patients after exhaustive exercise, an intraluminal high concentration of urates in the proximal straight tubule and/or thick ascending limb of Henle’s loop might stimulate the luminal Toll-like receptor 4–myeloid differentiation factor 88–phosphoinositide 3-kinase–mammalian target of rapamycin (luminal TLR4–MyD88–PI3K–mTOR) pathway to activate the nucleotide-binding oligomerization domain-like receptor family pyrin domain-containing 3 (NLRP3) inflammasome and may release interleukin-1β (IL-1β), which might cause the symptoms of EIAKI.

## 1. Introduction

Renal hypouricemia (RHUC) is a hereditary disease that presents with increased renal urate clearance and hypouricemia due to genetic mutations in the urate transporter that reabsorbs urates in the renal proximal tubule [1]. It is classified into the following two types:

Renal hypouricemia type 1 (RHUC1): A loss of function of URAT1, a transporter that uptakes urates into proximal tubule cells on the luminal side and increases renal urate clearance [2]. As the urate clearance/creatinine clearance ratio (FEua: fractional excretion of urate) is often below 100% in RHUC1 patients, it is thought there is another unidentified urate transporter on the luminal side in addition to URAT1.

Renal hypouricemia type 2 (RHUC2): The renal urate clearance increases due to a decrease in the function of GLUT9, a transporter that excretes urates at the basal side of the proximal tubule cells to the interstitium [3]. As the urate clearance/creatinine clearance ratio of RHUC2 patients often exceeds 100% and urate is predominantly secreted, it is thought that only GLUT9 is responsible for the exit of urate reabsorption on the basal side.

Exercise-induced acute kidney injury (EIAKI) is a complication of renal hypouricemia. Both renal hypouricemia types 1 and 2 have been reported to develop EIAKI [4,5]. After several hours of exhaustive (anaerobic) exercise, loin pain, nausea and vomiting develop and serum creatinine rises a few days later as signs and symptoms of EIAKI. The change of serum creatinine varies from a mild elevation to a severe elevation requiring dialysis. Delayed computed tomography (CT) scans after the administration of contrast media have demonstrated patchy wedge-shaped enhancements suggesting renal damage due to a vascular mechanism [6]. A renal biopsy may or may not show acute tubular necrosis. Although urate crystals are rarely found by a renal biopsy, it cannot be ruled out that renal biopsies are performed too late to detect them.

The mechanism of EIAKI should be referred to the case in Yeun and Hasbargen’s study [7]. A 29-year-old man with a history of recurrent EIAKI presented with hypouricemia with serum urate levels of 0.1–0.6 mg/dL and FEua of 69.2–383%. The pyrazinamide loading test reduced the FEua in this patient. Therefore, URAT1, the target molecule of pyrazinoate (a metabolite of pyrazinamide) was not deficient and he was not considered to be an RHUC1 patient. As the FEua exceeded 100%, it could be inferred that he was an RHUC2 patient. The probenecid loading test did not increase but decreased the FEua in this patient. It was considered that probenecid might inhibit the urate uptake by the OAT1 [8] and OAT3 [9] transporters, which are the entrances of urate secretion on the basal side of the proximal tubule, resulting in the suppression of urate secretion.

This patient and 4 healthy controls were subjected to exhaustive exercise and urine was collected at 1 h, 4 h and 24 h after the exercise to examine the urinary urate excretion. The patient’s creatinine clearance decreased from 151 mL/min before exercise to 77 mL/min by 4 h and 46 mL/min by 24 h after exercise, leading to acute kidney injury. The urate excretion/creatinine excretion ratio increased significantly from 0.29 before exercise to 1.49 at 4 h after exercise. The urate excretion/creatinine excretion ratio in the healthy control group remained almost unchanged from 0.30 ± 0.04 before exercise to 0.36 ± 0.04 at 4 h after exercise [7].

When all five subjects were administered 300 mg of allopurinol daily for 5 days and then re-exercised, the patient no longer had an acute kidney injury and the urate excretion/creatinine excretion ratio remained unchanged from 0.25 before exercise to 0.17 after exercise. Therefore, the authors concluded that an increased urinary urate excretion due to exhaustive exercise was the cause of EIAKI associated with renal hypouricemia and EIAKI could be prevented with the pre-administration of allopurinol [7].

Recently, we established new RHUC1 model mice that had a high activity of hypoxanthine phosphoribosyl transferase (HPRT), a loss of URAT1 transport function (Urat1KO) and a loss of uricase activity (UoxKO). We performed exhaustive exercise loading by forced swimming with these RHUC1 model mice (high HPRT-Urat1KO-UoxKO mice) and compared them with control mice (high HPRT-UoxKO mice). After exhaustive exercise, only the RHUC1 model mice demonstrated an increase in urinary urate excretion and plasma creatinine levels. The pro-IL-1β expression only increased in the post-exercise kidneys of the RHUC1 model mice. The increase in urate excretion after exercise disappeared with the administration of the xanthine oxidoreductase (XOR) inhibitor allopurinol or topiroxostat and the urates were excreted as hypoxanthine and xanthine. The intrarenal pro-IL-1β and plasma creatinine levels did not increase either (Figure 1) [10]. Therefore, we observed the same results as in Yeun and Hasbargen’s study using our RHUC1 model mice.

Our results reconfirmed that an increased urinary urate excretion after exhaustive exercise may be the cause of EIAKI associated with RHUC1. It was also newly revealed that an increased urinary urate excretion may cause EIAKI through renal pro-IL-1β production in patients with RHUC1.

## 2. EIAKI Due to Increased Urate Excretion after Exhaustive Exercise

Multiple cases of EIAKI have been reported in both types 1 and 2 of renal hypouricemia; there is no significant difference in the degree of renal damage between types 1 and 2. The inhibition of an increased urinary urate excretion after exercise suppressed the onset of EIAKI in the patient with renal hypouricemia type 2 in Yeun and Hasbargen’s study and in our RHUC1 model mice. Therefore, it can be assumed that the increase in urinary urate excretion after exercise is the cause of the onset of EIAKI in both type 1 and type 2 patients with renal hypouricemia.

As RHUC type 1 is a defect in the urate uptake on the luminal side of the proximal tubule, the intracellular urate concentration in the proximal tubule cells after exercise may be lower in RHUC1 patients than in healthy subjects. As RHUC type 2 has a deficiency in the urate excretion on the basal side of the proximal tubule, the intracellular urate concentration in the proximal tubule cells after exercise may be higher in RHUC type 2 patients than in healthy subjects (Figure 2). Assuming that the mechanism of EIAKI is the same for renal hypouricemia types 1 and 2, the intracellular compartment of proximal tubule cells is unlikely to be considered to be a lesion site in terms of the intracellular urate concentration.

What is common to renal hypouricemia types 1 and 2 is impaired urate reabsorption in the proximal tubules. In the downstream nephron segments where the intraluminal urate concentration should be low in normal kidneys, the intraluminal urate concentration becomes high due to the impaired upstream urate reabsorption in the kidneys of RHUC patients. In a micropuncture study of the kidneys of Cebus monkeys [11], the intraluminal urate concentration/plasma urate concentration ratio decreased to 0.71 ± 0.09 in the late proximal convoluted tubule and to 0.35 ± 0.05 in the early distal tubule. The intraluminal inulin concentration/plasma inulin concentration ratio increased to 2.7 ± 0.1 in the late proximal tubule and to 4.0 ± 0.2 in the early distal tubule (Figure 2). Therefore, the FEua decreased to 0.27 ± 0.03 in the late proximal convoluted tubule and to 0.09 ± 0.01 in the early distal tubule. A total of 73% of urates in the glomerular filtrate were reabsorbed in the proximal convoluted tubule, and an additional 18% were reabsorbed in the proximal straight tubule.

Assuming that no urates are reabsorbed in the proximal tubule of RHUC2 patients, the intraluminal urate concentration at the thick ascending limb of Henle’s loop should reach about three times the blood urate concentration, as same as the intraluminal concentration of creatinine. As the serum urate level was 4.7 mg/dL when the serum creatinine level was 5.1 mg/dL in a renal hypouricemia patient [4], the intraluminal urate concentration of the thick ascending limb reached 18.8 mg/dL. In the case of RHUC1, the reabsorption of urates other than URAT1 is thought to reduce the intraluminal urate concentration of the thick ascending limb.

In 2015, 100 µg/mL (10 mg/dL) of soluble urates were reported to stimulate Toll-like receptor 4 (TLR4) to enhance the expression of TLR4 and nucleotide-binding oligomerization domain (NOD)-like receptor (NLR) family pyrin domain-containing 3 (NLRP3) inflammasomes. The stimulated TLR4 also increased the production of caspase 1, interleukin-1β (IL-1β), and intercellular adhesion molecule 1 (ICAM-1) in cultured human proximal tubule cells [12]. These changes were suppressed by the TLR4 inhibitor TAK242, indicating that they were mediated by TLR4 in vitro. Therefore, it was revealed that extracellular soluble urates are a ligand for TLR4.

## 3. TLR4 in the Kidney

TLR4 is a Toll-like receptor expressed on the cell membrane surface and a typical ligand is the lipopolysaccharide (LPS) of Gram-negative bacteria [13]. In rat kidneys, the TLR4 molecule was reported to co-express constitutively with the Tamm–Horsfall protein in the thick ascending limb [14]. In the kidneys of mice, the expression of TLR4 protein was reported to be on both the luminal and the basal side of the medullary thick ascending limb [15]. However, another report demonstrated the expression of a TLR4 molecule on the luminal side of the proximal straight tubule (S3 segment) in mouse kidneys using a different anti-TLR4 antibody from the previous two studies [16]. Therefore, TLR4 was expressed constitutively at the luminal membrane of the proximal straight tubule and/or the thick ascending limb in the kidneys.

The stimulation of the luminal TLR4 of the medullary thick ascending limb of Henle with LPS caused a signal transduction via myeloid differentiation factor 88 (MyD88)–phosphoinositide 3-kinase (PI3K)–Akt serine/threonine kinase (Akt)–mammalian target of rapamycin complex 1 (mTORC1) and suppressed the HCO_3_^−^ reabsorption of this segment [17]. It has been reported that the stimulation of the basal TLR4 by LPS causes signal transduction via MyD88-IL-1 receptor-associated kinase 1 (IRAK1)-extracellular signal-regulated kinase (ERK) and reduced HCO_3_^−^ reabsorption [15]. Therefore, when the intraluminal urate concentration increases after exhaustive exercise, it is considered that the MyD88-PI3K-Akt-mTORC1 pathway is activated in the cells of the proximal straight tubules and/or the thick ascending limb where the TLR4 molecule is expressed constitutively on the luminal side.

Interleukin-1β (IL-1β) is produced by the activation of the NLRP3 inflammasome [18]. The activation of the NLRP3 inflammasome requires two steps: priming and triggering. The priming step is the expression of pro-IL-1β and NLRP3 through regulating the TLR4–inhibitor of NF-κB (IκB)–nuclear factor-κB (NF-κB) signaling and the phosphorylation of NLRP3 protein, which leads to its deubiquitination or ubiquitination of apoptosis-associated speck-like protein containing a CARD (ASC) protein to promote the assembly of NLRP3-ASC. In tubule cells, the stimulation of luminal and basal TLR4 with high concentration soluble urates after exercise may cause the expression in the priming step via NF-κB. The phosphorylation in the priming step may be caused by the basal TLR4–MyD88–IRAK1–TNF receptor-associated factor 6 (TRAF6)–TGF-β-activated kinase 1 (TAK1)–c-Jun N-terminal kinase 1 (JNK1) pathway [19,20] (Figure 3). As the basal TLR4–MyD88–IRAK1–ERK pathway was suppressed by the pre-treatment of the basal TLR4–Toll/IL-1 receptor domain-containing adaptor-inducing IFN-β (TRIF)–PI3K–Toll-interacting protein (Tollip) [21], hypouricemia of RHUC patients may attenuate the suppression via the basal TLR4–TRIF–PI3K–Tollip pathway and may enhance the phosphorylation in the priming step of the NLRP3 inflammasome activation via the basal TLR4–MyD88–IRAK1–TRAF6–TAK1–JNK1 pathway.

The triggering step of the NLRP3 inflammasome activation is required to form the inflammasome with NLRP3-ASC and pro-caspase-1. Stimulated luminal TLR4 also activates MyD88–PI3K–mTORC2–serum- and glucocorticoid-inducible kinase 1 (SGK1) and causes K^+^ efflux due to the activation of the renal outer medulla K^+^ channel (ROMK) [22]. NIMA-related kinase 7 (NEK7) detects K^+^ efflux and binds to the NLRP3 protein. NEK-bound NLRP3-ASC forms the NLRP3 inflammasome with pro-caspase-1 [23]. Therefore, K^+^ efflux and the drop of intracellular K^+^ concentration is a triggering signal for NLRP3 activation. (Figure 3).

Upon activation of the NLRP3 inflammasome, pro-caspase-1 is auto-cleaved and converted to caspase-1. Caspase-1 produces inflammatory cytokines from pro-IL-1β and pro-IL-18 to matured-form IL-1β and IL-18. It also produces gasdermin D N-terminal (GSDMD-NT) from gasdermin D (GSDMD), causing membrane permeabilization and inflammation-induced cell death (pyroptosis), which would be observed pathologically as acute tubular necrosis [24].

In our exercise loading study using RHUC1 model mice, the expression of pro-IL-1β was only enhanced in the post-exercise kidneys of the RHUC1 model mice but not in those of the control high HPRT-UoxKO mice. Through the component of the NLRP3 inflammasome, NLRP3, ASC and pro-caspase-1 were expressed both in the pre-exercise kidneys of the RHUC1 model mice and the control mice. It has also been shown that the activation of mTORC1 can promote the expression and maturation of IL-1β through hypoxia-inducible factor-1α (HIF-1α) [25,26]. As the stimulation of the luminal TLR4 of the medullary thick ascending limb caused the signal transduction via MyD88–PI3K–Akt–mTORC1 [17], it was considered that an intraluminal high urate concentration might enhance the expression of pro-IL-1β only in the post-exercise kidneys of RHUC1 model mice (Figure 3). In the future study, we will have to identify the nephron segments where pro-IL-1β will be expressed after an exhaustive exercise load in the kidneys of RHUC1 model mice.

## 4. IL-1β Production and Acute Kidney Injury

Nausea, vomiting and loin pain are almost inevitable symptoms in the onset of EIAKI associated with renal hypouricemia [6]. IL-1β has the effect of activating the sensory nerves, and the activation of sensory nerves has been reported even at low concentrations of 10^−18^ M [27]. The distribution of sensory nerves in the kidneys is predominant in the renal pelvis, but they are also partially distributed around the cortical tubules [28]. The ascending nerves from the kidneys pass through the spinal cord to the solitary tract nucleus of the medulla oblongata (which is involved in nausea and vomiting [29]) to the cranial ventrolateral medulla oblongata (which is the center of the sympathetic nerves) and to the paraventricular nucleus. Therefore, increased post-exercise urinary urate excretion in renal hypouricemia patients induces IL-1β production in renal tubule cells and IL-1β may stimulate the sensory nerves around the cortical tubules to stimulate the medulla oblongata nucleus and cause nausea and vomiting. In addition, it has been considered that IL-1β may stimulate the nociceptive nerve to cause loin pain [30].

The stimulation of the ventrolateral medulla oblongata and paraventricular nucleus of the hypothalamus is thought to stimulate the sympathetic nerve activity in the kidneys as a renorenal reflex. The sympathetic nerve is thought to constrict afferent arterioles in the kidneys and decrease GFR [28]. It has been indicated in the involvement of sympathetic nerve activity that etilefrine, an αβ agonist, induced an acute kidney injury in a renal hypouricemia patient without exhaustive exercise [31].

## 5. Increased Urinary Urate Excretion Due to Exercise

ATP utilization (ATP + H_2_O => ADP + P_i_ + H^+^) underpins skeletal muscle contraction during exercise. Because the intramuscular stores of ATP are relatively small, oxidative phosphorylation (aerobic metabolism pathways) indicated below must be activated for ATP resynthesis in mitochondria.
Glucose + 6 O_2_ + 36 ADP => 6 CO_2_ + 6 H_2_O + 36 ATP
Palmitate + 23 O_2_ + 130 ADP => 16 CO_2_ + 16 H_2_O + 130 ATP

Aerobic exercises like jogging utilize ATP resynthesized mainly by aerobic metabolism pathways. When the intensity of exercise increased and the oxygen supply to skeletal muscle became short of oxygen demand for aerobic metabolism pathways, anaerobic metabolism pathways indicated below were activated by accumulated ADP.
Phospho-creatine + ADP + H^+^ => ATP + creatine
2 ADP => ATP + AMP
Glucose (or glycogen _n_) + 3 ADP + 3 P_i_ => (glycogen _n__−1_) + 2 lactate + 3 ATP

Anaerobic exercises like sprinting utilize ATP resynthesized partly by anaerobic metabolism pathways [32].

In anaerobic exercise, the accumulated ADP also activates adenylate kinase in the skeletal muscle cells, producing one molecule of ATP and one molecule of AMP from two molecules of ADP [33]. AMP is converted to inosine 1-phosphate by AMP deaminase and decomposed to hypoxanthine via inosine [34]. As there is almost no xanthine oxidoreductase activity in skeletal muscle cells [35], intracellular hypoxanthine is released into the blood. Hypoxanthine in the blood is taken up by the liver, converted to urates and released into the blood [36]. Urates released into the blood are excreted in the urine.

During exercise, the oxygen supply to the skeletal muscles rises to meet the increasing demand for oxygen. An increased blood flow (hyperemia) in the skeletal muscle is triggered by vasodilators locally produced in the muscle tissue, either intraluminally or extraluminally of the blood vessels. Although many vasodilators have been shown to provide this increase in blood flow, nitric oxide (NO) and prostacyclin are important. The production of both vasodilators is stimulated by adenosine, ATP, acetylcholine and bradykinin. It is also affected by mechanical stimuli such as shear stress. NO and prostacyclin have also been shown to interact in a complementary manner so that when one system is compromised, the other system can supplement it. Most exercise hyperemia remains unexplained. Unexplained exercise hyperemia may be explained by cAMP and cGMP-independent smooth muscle relaxation such as endothelium-derived hyperpolarizing factor (EDHF) or metabolic sympathetic regulation [37].

The classic EDHF pathway involves the hyperpolarization of the vascular endothelial cell secondary to an increase of intracellular Ca^2+^, which leads to the K^+^ channel activation. Vascular smooth muscle cell hyperpolarization occurs as a result of K^+^ channel activation by extracellular K^+^ from the endothelial cells or by electrical transmission through gap junctions between endothelial cells and smooth muscle cells. The non-classic EDHF pathway does not require the hyperpolarization of endothelial cells but requires endothelial-derived factors (H_2_O_2_, NO, prostaglandin and epoxyeicosatrienoic acids (EET)) to act on the K^+^ channels or transporters of the smooth muscle cells to induce hyperpolarization and cell relaxation [38].

## 6. Post-Exercise Urate Production in Patients with Renal Hypouricemia

In the patients with renal hypouricemia in Yeun and Hasbargen’s study, the urate excretion/creatinine excretion ratio (mg dL^−1^/mg dL^−1^) increased significantly from 0.29 before loading to 1.49 at 4 h after loading. In the healthy subjects, there was almost no change at 0.30 ± 0.04 before loading and 0.36 ± 0.04 at 4 h after loading [7]. The urate excretion/creatinine excretion ratio (µmol/mg) in the healthy subjects in another study decreased from 0.4 before loading at 1 h after exhaustive exercise, then increased to 0.6 at 2 h after exercise and remained up to 4 h after exercise [39]. As urinary allantoin increases during post-exercise urinary urate reduction, it is possible that a proportion of the urates to be excreted may be non-enzymatically converted to allantoin. It is thought that urates in the blood enter the skeletal muscle and eliminate the active oxygen species in the skeletal muscles to be converted to allantoin [40].

An increased post-exercise urate excretion in patients with renal hypouricemia can be interpreted as an excretion of urates that were not reabsorbed in the kidneys. The question is then if exercise-induced hypoxanthine production is the same in healthy individuals and RHUC patients.

An impaired flow-mediated dilation was observed in RHUC1 patients with a urate concentration below 47.6 μmol/L (<0.8 mg/dL). No expression of URAT1 was demonstrated in cultured human umbilical vascular endothelial cells (HUVEC) [41]. This study indicated that the endothelial dysfunction in RHUC patients was not caused by a URAT1 defect in the endothelial cells but by a low serum urate level.

A randomized, double-blind crossover study of 17 healthy young men was reported using a placebo, febuxostat alone and a combination of rasburicase and febuxostat. Extremely low concentrations of urates did not alter acetylcholine and sodium nitroprusside-induced vasodilation but reduced the heat-induced non-NO-mediated late thermal vasodilation response after the iontophoresis of the NO synthase inhibitor. An increased serum EET in a low concentration of urates was reversed in extremely low concentrations of serum urates. Therefore, an extremely low concentration of serum urates might disturb the EET-related EDHF pathway vasodilation response to local heating [42].

Based on the above, EDHF is involved in exercise hyperemia, and EDHF-dependent vasodilation is inhibited in hypouricemia. Therefore, exercise hyperemia may be disturbed in RHUC patients. As a hypothesis, assuming that exercise hyperemia is disturbed in RHUC patients compared with healthy subjects, ATP regeneration by oxidative phosphorylation may be disturbed even with the same intensity of exercise, and ATP degradation may be accelerated. Urinary urate excretion due to exhaustive exercise may be increased in renal hypouricemia compared with healthy subjects. To test this hypothesis, we will have to examine the changes in the total urinary urate excretion with an exhaustive exercise load in our RHUC1 model mice.

In conclusion, in the skeletal muscle of renal hypouricemia patients during exhaustive exercise, the decreased endothelial EET release by hypouricemia might cause the disturbance of exercise hyperemia, which increases the post-exercise urinary urate excretion. In the kidneys of renal hypouricemia patients after exhaustive exercise, an intraluminal high concentration of urates in the proximal straight tubule and/or the thick ascending limb of Henle might stimulate the TLR4–MyD88–PI3K–mTOR pathway and release IL-1β, which might cause symptoms of EIAKI associated with renal hypouricemia. These hypothetical mechanisms must be tested in the future.

## Figures and Tables

**Figure 1 biomedicines-09-01847-f001:**
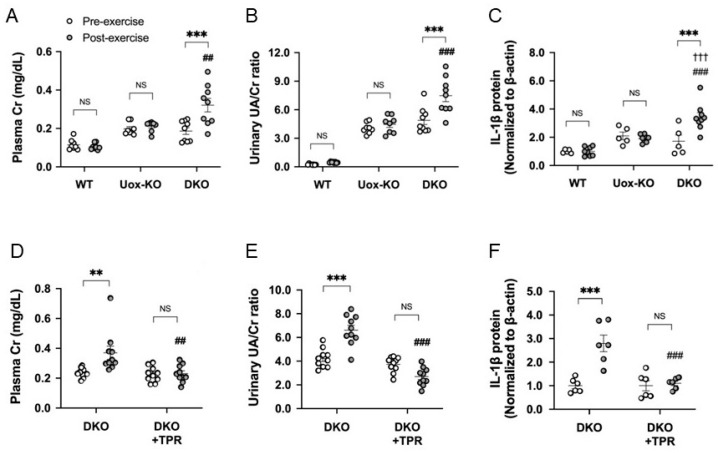
EIAKI of high HPRT activity Urat1KO-UoxKO mice and effects of topiroxostat. (**A**) Plasma Cr (mg/dL), (**B**) urinary UA/Cr ratio, (**C**) pro-IL-1β in kidney of WT (B6), high HPRT activity UoxKO (Uox-KO) and high HPRT activity Urat1KO-UoxKO (DKO) mice at pre- (open circles) and post-exercise (dark circles) state. (**D**) Plasma Cr (mg/dL), (**E**) urinary UA/Cr ratio, (**F**) pro-IL-1β in the kidney of the DKO control and topiroxostat-treated DKO (DKO+TPR) mice. Significant difference from pre-exercise group (** *p* < 0.01, *** *p* < 0.001), WT mice group (††† *p* < 0.001) and Uox-KO group (A–C: ## *p* < 0.01, ### *p* < 0.001) or DKO control group (D–F: ## *p* < 0.01, ### *p* < 0.001). NS, not significant (*p* > 0.05). Figures are from the reference [10].

**Figure 2 biomedicines-09-01847-f002:**
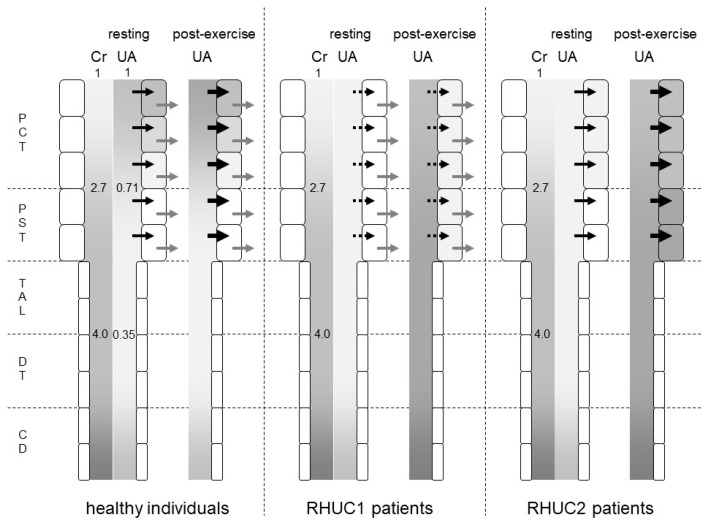
Predicted intraluminal/plasma concentration ratio of creatinine and urates along the nephron segments of healthy individuals, RHUC1 and RHUC2 patients at resting or post-exercise state. The gradation represents the intraluminal/plasma concentration ratio of creatinine (Cr) and urates (UAs). The darker gradation indicates a higher intraluminal/plasma concentration ratio. The intraluminal/plasma concentration ratio of Cr was predicted to be 2.7 and 4.0 at the late proximal convoluted tubule (PCT) and early distal tubule (DT), respectively. In healthy individuals, the intraluminal/plasma concentration ratio of UA was predicted to be 0.71 and 0.35 at the late PCT and early DT, respectively. Black and gray arrows represent the urate influx via URAT1 and the urate efflux via GLUT9, respectively. Large black arrows at post-exercise state indicate the enhanced urate influx via URAT1 by lactate. Dashed black arrows indicate a small urate influx via another unidentified urate transporter. At a post-exercise state, the plasma urate concentration of RHUC patients increased to reach that of healthy individuals at resting state. The darker gradation indicates that the intraluminal/plasma UA concentration ratio at the proximal straight tubule (PST) and the thick ascending limb (TAL) in RHUC patients should be higher than those in healthy individuals. The intracellular urate concentration of the proximal tubule cells in RHUC2 patients was predicted to be higher than that in RHUC1 patients. CD: collecting duct.

**Figure 3 biomedicines-09-01847-f003:**
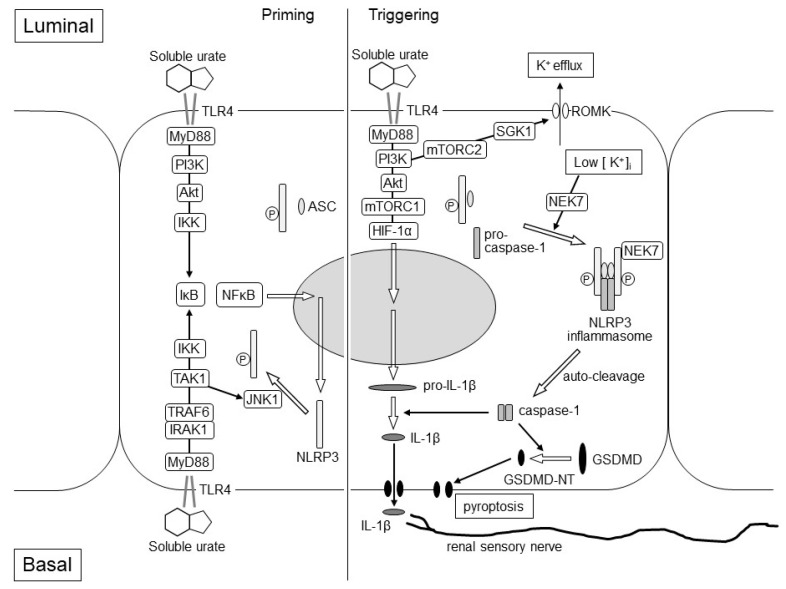
Hypothetical Mechanism of EIAKI in the kidneys of RHUC patients. The figure represents assumed intracellular events after TLR4 stimulation of intraluminal and basal soluble urates, which elevated transiently at the post-exhaustive exercise state of RHUC patients. The left and right parts of the figure represent the priming and triggering steps of the NLRP3 inflammasome activation, respectively. In the priming step, the expression of NLRP3 protein may be enhanced via the luminal TLR4–MyD88–PI3K–Akt–IKK–IκB–NF-κB pathway and/or the basal TLR4–MyD88–IRAK1–TRAF6–IKK–IκB–NF-κB pathway. The phosphorylation of the NLRP3 protein may be induced via the basal TLR4–MyD88–IRAK1–TRAF6–TAK1–JNK1 pathway to promote the assembly of NLRP3-ASC. In the triggering step, the NEK7-dependent assembly of the NLRP3 inflammasome with NLRP3-ASC and pro-caspase-1 may be stimulated by the luminal TLR4-MyD88-PI3K-mTORC2-SGK1 pathway, which causes K^+^ efflux due to the activation of the ROMK channel. Luminal TLR4 may also promote the expression of the pro-IL-1β via MyD88-PI3K-Akt-mTORC1-HIF-1α pathway. Pro-caspase-1 is auto-cleaved and converted to caspase-1, which produces matured IL-1β and GSDMD-NT from pro-IL-1β and GSDMD, respectively. IL-1β stimulates the renal sensory nerve endings around the cortical tubule, which may cause nausea, vomiting, loin pain and afferent arteriole constriction. GSDMD-NT causes membrane permeabilization and proptosis. Akt: Akt serine/threonine kinase; ASC: apoptosis-associated speck-like protein containing a CARD protein; GSDMD-NT: gasdermin D N-terminal; HIF-1α: hypoxia-inducible factor-1α; IκB: inhibitor of NF-κB; IKK: IκB kinase; IL-1β: interleukin-1β; IRAK1: IL-1 receptor-associated kinase 1; JNK1: c-Jun N-terminal kinase 1; mTORC1: mammalian target of rapamycin complex 1; mTORC2: mammalian target of rapamycin complex 2; MyD88: myeloid differentiation factor 88; NEK7: NIMA-related kinase 7; NF-κB: nuclear factor-κB; NLRP3: nucleotide-binding oligomerization domain-like receptor family pyrin domain-containing 3; PI3K: phosphoinositide 3-kinase; ROMK: renal outer medulla K^+^ channel; SGK1: serum and glucocorticoid-inducible kinase 1; TAK1: TGF-β-activated kinase 1; TLR4: Toll-like receptor 4; TRAF6: TNF receptor-associated factor 6.

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
