# Peer review of "Hypothetical Mechanism of Exercise-Induced Acute Kidney Injury Associated with Renal Hypouricemia"

_biomedicines, 2021, doi:10.3390/biomedicines9121847_

Round 1
Reviewer 1 Report
The manuscript presented for review presents inquiries related to the development of exercise-induced acute kidney injury associated with renal hyperuricemia. The article contains very interesting, well-described and important issues related to the mechanisms leading to the occurrence of kidney damage in cases that are the subject of the publication. I have no criticisms of the content presented in the manuscript and believe that it may be published as is. I only noticed a minor typo, line 22 - "patient" instead of "patients".
Author Response
Response to Reviewer 1
Comments and Suggestions for Authors
The manuscript presented for review presents inquiries related to the development of exercise-induced acute kidney injury associated with renal hyperuricemia. The article contains very interesting, well-described and important issues related to the mechanisms leading to the occurrence of kidney damage in cases that are the subject of the publication. I have no criticisms of the content presented in the manuscript and believe that it may be published as is. I only noticed a minor typo, line 22 - "patient" instead of "patients".
→ Thank you for your comment. I have corrected the typo (page 2, line 22).
Reviewer 2 Report
Hosoyamada wrote a review manuscript regarding hypothetical mechanism of exercise-induced acute kidney injury associated with renal hypouricemia. Referring to the previous publications regarding the case in Yeun and Hasbargen’s study, severe hypouricemia and endothelium-derived vasodilation in healthy young men in De Becker’s study, RHUC1 model mice and other basic studies regarding TLR4 signaling and NLRP3 activation, the author attempted to construct the theory about the hypothetical mechanisms of exercise-induced acute kidney injury associated with renal hypouricemia.
General comments
This review is full of updated information from recent clinical and basic scientific publications, and comprehensive, educative and help the researchers in this area organize the knowledge about RHUC. The author needs to respond to comments raised as below to polish this review before publication.
Major comments
- I assume that author intended to title ‘Hypothetical mechanism of exercise-induced acute kidney injury associated renal hypouricemia’, instead of acute kidney injury associated with renal hyperuricemia.
- In Figure 1 on page 3, please consider to describe the figures of wild type with post-exhaustive exercise and/or the figures of RHUC1/2 in normal resting condition, which would be more helpful to understand to readers.
- On page 4, TLR4 in the Kidney part, the author discussed that intraluminal increased urate stimulate TLR4 as LPS. Is there any scientific evidence that intraluminal urate does stimulate the TLR4 expressed in proximal straight tubule and/or the thick ascending limb in human kidneys in vivo?
4) On page 4, TLR4 in the Kidney part. What is the difference between increased intraluminal urate in RHUC1 patients after exhaustive exercise and that in gout patients treated with uricouric stimulant after exhaustive exercise? Dose increased intraluminal urate in gout patients treated with uricosuric stimulant stimulate TLR4 similarly after exhaustive exercise? If not, intraluminal increased urate is not only factor to induce acute kidney injury and please explain the other assumed mechanisms than intraluminal urate levels to cause acute kidney injury in RHUC1 patients after exhaustive exercise (such as the possible difference of TLR4 expression levels…)
5) In line 33, on page 5, please showed the data in that model if available for readers to understand deeply.
Minor comments
- In line 34 on page 2, the typo RHIC1 should be corrected to RHUC1.
- In line 13 on page 5, the word ‘serum’ should be deleted.
- In line 15 on page 5, the sentence, resulting in the activation of the NLRP3 inflammasome to sense the K+ efflux by NIMA-related kinase 7 (NEK7). The Figure 2 showed that NEK7 activated by sensing K+ is activating NLRP3. However, some readers might be confused, since the sentence above might means that activated NLRP3 inflammasome makes NEK7 sense K+ efflux. Please explain the relationship of K+ efflux, NEK7 and NLRP3 inflammasome more clearly in other sentences.
4) The title of figure 2, Mechanism of EIAKI in the kidneys of RHUC patients. Almost all pathways depicted in this figure were tested in the animal models and we are not still confident with the completely same pathways employed in the RHUC patients and potentially would mislead the readers and researchers in this area. Please modify the title.
Author Response
Response to Reviewer 2
Comments and Suggestions for Authors
Hosoyamada wrote a review manuscript regarding hypothetical mechanism of exercise-induced acute kidney injury associated with renal hypouricemia. Referring to the previous publications regarding the case in Yeun and Hasbargen’s study, severe hypouricemia and endothelium-derived vasodilation in healthy young men in De Becker’s study, RHUC1 model mice and other basic studies regarding TLR4 signaling and NLRP3 activation, the author attempted to construct the theory about the hypothetical mechanisms of exercise-induced acute kidney injury associated with renal hypouricemia.
General comments
This review is full of updated information from recent clinical and basic scientific publications, and comprehensive, educative and help the researchers in this area organize the knowledge about RHUC. The author needs to respond to comments raised as below to polish this review before publication.
Major comments
- I assume that author intended to title ‘Hypothetical mechanism of exercise-induced acute kidney injury associated renal hypouricemia’, instead of acute kidney injury associated with renal hyperuricemia.
→ Thank you very much for your comment. I have corrected my terrible mistake in the title (page 1, line 3).
- In Figure 1 on page 3, please consider to describe the figures of wild type with post-exhaustive exercise and/or the figures of RHUC1/2 in normal resting condition, which would be more helpful to understand to readers.
→ Thank you for your suggestion. I have revised old Figure 1 as new Figure 2 (page 4).
- On page 4, TLR4 in the Kidney part, the author discussed that intraluminal increased urate stimulate TLR4 as LPS. Is there any scientific evidence that intraluminal urate does stimulate the TLR4 expressed in proximal straight tubule and/or the thick ascending limb in human kidneys in vivo?
→ Thank you for your comment. Unfortunately, there is no scientific evidence about EIAKI mechanism in the kidneys of RHUC patients now. Because no EIAKI models have demonstrated with RHUC model mice until our first report [10], this review is the first proposal of a hypothetical EIAKI mechanism in RHUC. We will have to test our hypothesis by pre-clinical and clinical studies in future.
- On page 4, TLR4 in the Kidney part. What is the difference between increased intraluminal urate in RHUC1 patients after exhaustive exercise and that in gout patients treated with uricosuric stimulant after exhaustive exercise? Dose increased intraluminal urate in gout patients treated with uricosuric stimulant stimulate TLR4 similarly after exhaustive exercise? If not, intraluminal increased urate is not only factor to induce acute kidney injury and please explain the other assumed mechanisms than intraluminal urate levels to cause acute kidney injury in RHUC1 patients after exhaustive exercise (such as the possible difference of TLR4 expression levels…)
→ Thank you for your significant comment. At first, the FEua increased by 30 mg/kg of dotinurad and benzbromarone were reported 25.2% and 12.4% (Cebus monkeys), respectively (indicated below). On the other hand, FEua of RHUC1 patients reported 58.4%. Therefore, the inhibition of urate reabsorption by uricosuric drugs may be incomplete to reach the same level of urate reabsorption in RHUC patients. Second, as reviewer suggested, it was reported that protection mechanism of basal TLR4 signaling by basal pretreatment of TLR4 agonist. Thus, basal TLR4 stimulation by hyperuricemia might attenuate luminal TLR4 signaling to EIAKI. Further studies will be required to answer this comment.
Taniguchi, T. et. al., Pharmacological Evaluation of Dotinurad, a Selective Urate Reabsorption Inhibitor. J Pharmacol Exp Ther 2019, 371 (1), 162-170.
- In line 33, on page 5, please showed the data in that model if available for readers to understand deeply.
→ Thank you for your comment. The indicated sentence becomes in line 14, on page 6 in the revised manuscript. However, the localization study of IL-1β protein in the nephron segment have not started yet. Instead of future result, I demonstrate EIAKI and effects of topiroxostat data of RHUC1 model mice as new Figure 1 (page 3, line 2) from the reference [10], which was published online on Nov. 19th.
Minor comments
- In line 34 on page 2, the typo RHIC1 should be corrected to RHUC1.
→ Thank you for your comment. I have corrected (page 2, line 34).
- In line 13 on page 5, the word ‘serum’ should be deleted.
→ Thank you for your comment. A hyphen had to be added after ‘serum’. The phrase was corrected as ‘serum- and glucocorticoid-inducible kinase 1 (SGK1)’(page 5, line 42).
- In line 15 on page 5, the sentence, resulting in the activation of the NLRP3 inflammasome to sense the K+ efflux by NIMA-related kinase 7 (NEK7). The Figure 2 showed that NEK7 activated by sensing K+ is activating NLRP3. However, some readers might be confused, since the sentence above might means that activated NLRP3 inflammasome makes NEK7 sense K+ Please explain the relationship of K+ efflux, NEK7 and NLRP3 inflammasome more clearly in other sentences.
→ Thank you for your suggestion. I have added new sentences as follows: “Stimulated luminal TLR4 also activates MyD88PI3KmTORC2serum- and glucocorticoid-inducible kinase 1 (SGK1) and causes K+ efflux due to the activation of the renal outer medulla K+ channel (ROMK) [22]. NIMA-related kinase 7 (NEK7) detects K+ efflux and binds to the NLRP3 protein. NEK-bound NLRP3-ASC forms the NLRP3 inflammasome with pro-caspase-1 [23]. Therefore, K+ efflux and the drop of intracellular K+ concentration is a triggering signal for NLRP3 activation. (Figure 3) ” (page 5, line 44-47).
And old Figure 2 was modified to new Figure 3, in which NEK7 protein binds to NLRP3 protein in NLRP3 inflammasome.
- The title of figure 2, Mechanism of EIAKI in the kidneys of RHUC patients. Almost all pathways depicted in this figure were tested in the animal models and we are not still confident with the completely same pathways employed in the RHUC patients and potentially would mislead the readers and researchers in this area. Please modify the title.
→ Thank you for your comment. I have revised the title of new Figure 3 as “Hypothetical Mechanism of EIAKI in the kidneys of RHUC patients”(page 6, line 17).

Reviewer 3 Report
Dear Editor,
Enclosed please find the comments on the following manuscript:
Manuscript ID: biomedicines- 1466288
Type of manuscript: Review
Title: Hypothetical Mechanism of Exercise-Induced Acute Kidney Injury Associated with Renal Hyperuricemia
This review article demonstrates the hypothetical mechanism of kidney disorder between exercise and renal hyperuricemia. In my opinion, there are major concerns that impair the publication of this review in the present form.
- Authors should focus on what kind of exercise will cause acute kidney injury. Please let readers understand the definition between aerobic and anaerobic exercise first. The introduction should provide enough background and include all relevant references.
- Some references are missing in this review article. Page 2, line 20 and line26: Authors should cite references. Page 4, line7: Authors should cite references.
- Page6, line 32: Authors should revise the format of 10-18.
- Page7, line14: The paragraph of Increased Urinary Urate Excretion Due to Exercise is not adequately described and clearly presented. Authors should organize this paragraph more carefully.
- Page 8, line 8-40: The conclusions lack solid evidence supported by the results.
Author Response
Reviewer 3
Comments and Suggestions for Authors
This review article demonstrates the hypothetical mechanism of kidney disorder between exercise and renal hyperuricemia. In my opinion, there are major concerns that impair the publication of this review in the present form.
- Authors should focus on what kind of exercise will cause acute kidney injury. Please let readers understand the definition between aerobic and anaerobic exercise first. The introduction should provide enough background and include all relevant references.
→ Thank you for your suggestion. I have revised as follows: “After several hours of exhaustive (anaerobic) exercise, loin pain, nausea and vomiting develop and serum creatinine rises a few days later as signs and symptoms of EIAKI”(page 1, line 39-41). The explanation of aerobic and anaerobic exercise has been added the later paragraph (page 7, line 30-44), which is required to be revised by #4 comment.
- Some references are missing in this review article. Page 2, line 20 and line26: Authors should cite references. Page 4, line7: Authors should cite references.
→ Thank you for your comment. They are the description from Yeun and Hasbargen's study as same as the previous paragraph. Thus, I have added the reference [7] at page 2, line 20 and line26.
Page 4, line7 in submitted old manuscript was the #9 font and the end of figure legend of old Figure 1. The page 4, line 10 in submitted old manuscript was #10 font and the start of main text. I have added space between as line 18-19 in revised new manuscript. The explanation of figure legend of new Figure 2 (corresponded to old Figure 1) was described in the main text (page 2, line 52 – page 3, line 31).
- Page6, line 32: Authors should revise the format of 10-18.
→ Thank you for your comment. “ 10-18 ” have been substituted for “ 10⁻¹⁸ ”(page 7, line 13).
- Page7, line14: The paragraph of Increased Urinary Urate Excretion Due to Exercise is not adequately described and clearly presented. Authors should organize this paragraph more carefully.
→ Thank you for your significant comment. I have divided the section of “Increased Urinary Urate Excretion Due to Exercise” into four paragraphs: Aerobic and anaerobic metabolism pathway with explanations of aerobic and anaerobic exercise; purine degradation in anaerobic metabolism pathway; hyperemia in the skeletal muscle during exercise; EDHF pathway as a candidate mechanism of unexplained exercise hyperemia (page 7, line 30 – page 8, line 21). Five references ([32]-[36]) have been added newly.
- Page 8, line 8-40: The conclusions lack solid evidence supported by the results.
→ Thank you for your comment. there is no solid evidence about EIAKI mechanism in the kidneys of RHUC patients now. Because no EIAKI models have demonstrated with RHUC model mice until our first report [10], this review is the first proposal of a hypothetical EIAKI mechanism in RHUC. We will have to test our hypothesis by pre-clinical and clinical studies in future. Thus, I have substituted “may” for “might” and added “These hypothetical mechanisms must be tested in the future” as the final sentence.

Round 2
Reviewer 3 Report
Dear Editor,
Enclosed please find the comments on the following manuscript:
Manuscript ID: biomedicines- 1466288
Type of manuscript: Review
Title: Hypothetical Mechanism of Exercise-Induced Acute Kidney Injury Associated with Renal Hypouricemia
I have read this review carefully and feel that this manuscript is suitable for publish in the present form.